# HYPOCRITE: HOMOGLYPH ADVERSARIAL EXAMPLES FOR NATURAL LANGUAGE WEB SERVICES IN THE PHYSICAL WORLD

## ABSTRACT

Recently, as Artificial Intelligence (AI) develops, many companies in various industries are trying to use AI by grafting it into their domains. Also, for these companies, various cloud companies (e.g., Amazon, Google, IBM, and Microsoft) are providing AI services as the form of Machine-Learning-as-a-Service (MLaaS). However, although these AI services are very advanced and well-made, security vulnerabilities such as adversarial examples still exist, which can interfere with normal AI services. This paper demonstrates a HYPOCRITE for hypocrisy that generates homoglyph adversarial examples for natural language web services in the physical world. This hypocrisy can disrupt normal AI services provided by the cloud companies. The key idea of HYPOCRITE is to replace English characters with other international characters that look similar to them in order to give the dataset noise to the AI engines. By using this key idea, parts of text can be appropriately replaced with subtext with malicious meaning through black-box attacks for natural language web services in order to cause misclassification. In order to show attack potential by HYPOCRITE, this paper implemented a framework that makes homoglyph adversarial examples for natural language web services in the physical world and evaluated the performance under various conditions. Through extensive experiments, it is shown that HYPOCRITE is more effective than other baseline in terms of both attack success rate and perturbed ratio.

## 1 INTRODUCTION

Artificial Intelligence (AI) has shown the potential of convenience in many domains. With the advance of AI, people are living affluent lives by AI. AI can judge what is difficult for humans to make, classify what humans struggle with, predict what humans can never measure, and even recommend tasks that fall within a pattern (Naumov et al. (2019)). Due to the development of the AI industry, it is not an exaggeration to say that mankind coexists with AI as many companies in various industries are trying to use AI by grafting it into their domains. As the demand on AI increases, various cloud service providers (e.g., Amazon Comprehend (Amazon), Google Cloud Natural Language (Google) , Watson Natural Language Understanding (IBM), and Text Analytics (Microsoft) are providing easy-to-use Machine-Learning-as-a-Service (Ribeiro et al. (2015)) to people and companies who want to use AI services through their cloud. Among the MLaaS, Natural Language Processing (NLP) based on text is one of the important AI services.

NLP, which contains various information such as emotional and semantic analysis of text-based data (Dang et al. (2020); Kamath & Ananthanarayana (2016)), can be used to develop platforms for various recommendation systems. For example, it is possible to provide effective data analysis to corporate management based on quick information delivery by identifying the needs of users and identifying only the core of a system or long article based on the sentiment analysis **?**sentiment-analysis) based on the user's review. Like this various cloud service providers' MLaaS superiority is sufficiently proven through many studies.

However, although these AI services are very advanced and well-made, security vulnerabilities are still existing. Because these security vulnerabilities can interfere with normal AI services, so cause a fatal problem, the integrity of such services should be protected. This paper shows the security

vulnerabilities for natural language web services in the physical world (Rodriguez et al. (2019)). The key idea of the adversarial examples( Goodfellow et al. (2014); Creswell et al. (2018)) for natural language web services is to replace English characters with other similar international characters (e.g., homoglyph) in order to give the dataset noise (Boucher et al. (2021)). By using this key idea, parts of text can be appropriately replaced with subtext with malicious meaning through black-box attacks(Ilyas et al. (2018)) for natural language web services in order to cause misclassification.

The main contributions of this paper are summarized as follows:

- **Text adversarial examples for natural language web services in the Physical World:** In order to show the feasibility of our attack, we implemented a framework that can generate text adversarial examples for natural language web services in the physical world (see Section 3).

- **Untargeted attacks and targeted attacks:** For various goals of the adversarial attacks, we carried out the text adversarial attacks for not only non-targeted attacks (i.e., misclassification) but also targeted attacks (i.e., targeted misclassification and source/target misclassification) (see Section 3).

- **The performance evaluation of the proposed framework:** Through extensive experiments, it is shown that the proposed framework outperforms a baseline framework in terms of both attack success rate and perturbed ratio (see Section 4).

- **The impact of human understanding:** To evaluate the attack text generated by our proposed adversarial attack model, we used Amazon Mechanical Turk (Mturk) how difficult it is to find the attacked word, we conducted a survey using the attack text and obtained and analyzed the success rate of the attack on the survey problem (see Section 4.2.2).

The remainder of this paper is organized as follows. The background and related work of text adversarial examples is given in Section 2. Section 3 describes the overview of the proposed adversarial attack and explains the process of the text adversarial attack of generating text adversarial examples for natural language web services in the physical world. Section 4 evaluates the performance of the our proposed framework through misclassification attacks for sentiment analysis of natural language web services in the physical world. Section 5 discusses some research challenges for our attack. Finally, Section 6 concludes this paper along with future work.

## 2   RELATED WORK

Research on Adversarial attacks for NLP models has been presented. In 2016, (Papernot et al. (2016)) proposes a method to craft a sequential input on Recurrent Neural Network (RNN) models to manipulate an output of classifiers. (Ebrahimi et al. (2017)) presents a method to generate adversarial examples for text classification by crafting a few characters of an input string. Unlike (Ebrahimi et al. (2017)) whose attacks were white-box adversarial examples, (Gao et al. (2018)) used black-box adversarial text sequence to make deep learning-based classifiers misclassify. (Li et al. (2018)) shows various methodologies to generate adversarial text for NLP models, and evaluates that popular NLP services for web services are vulnerable to those attacks. In our work, we show a new methodology to generate adversarial text which is not considered in (Li et al. (2018)), and also show that most of the NLP services are still vulnerable to our attack. (Wolff & Wolff (2020)) proposes homoglyph attacks generating adversarial examples to neural text detectors. Our attacks are targeted to sentimental analysis services in the real world, and we try to perturb every unit of target text (e.g., word, sentence, and paragraph) instead of replacing several letters with homoglyphs in order to show the most effective way to generate adversarial examples.

## 3   ATTACK MODEL

This section presents the goal, overview, and attack process of HYPOCRITE. The goal of HYPOCRITE is to generate text adversarial examples for a sentiment analysis of natural language web services in the physical world. In other words, the generated adversarial examples can cause misclassification from positive sentiment to negative sentiment or from negative sentiment to positive sentiment.

**Original Text**

This film is really bad, so bad that even Christopher Lee cannot save it. A poor story an even poorer script and just plain bad direction makes this a truly outstanding horror film, the outstanding part being that it is the only horror film that i can honestly say i would never ever watch again. This garbage make Plan nine from outerspace look like oscar material.

### Amazon Comprehend (Amazon)

**Original Result** : Negative (0.9986)     **Adversarial Example Result** : Positive (0.7041)

This film is really **bad**, **so bad** that even **Christopher Lee** cannot save it. A **poor** story an even **poorer** script and just plain **bad** direction makes this a truly outstanding **horror** film, the outstanding part being that it is the only **horror** film that i can honestly say i would never ever watch again. This **garbage** make Plan nine from outerspace look like oscar material.

### Google Cloud Natural Language AI (Google)

**Original Result** : Negative (-0.6)     **Adversarial Example Result** : Positive (0.4)

This film is really **bad**, so **bad** that even Christopher Lee cannot save it. A **poor** story an even **poorer** script and just plain **bad** direction makes this a truly outstanding horror film, the outstanding part being that it is the only horror film that i can honestly say i would never ever watch again. This **garbage** make Plan nine from outerspace look like oscar material.

### Watson Natural Language Understanding (IBM)

**Original Result** : Negative (0.877)     **Adversarial Example Result** : Positive (0.3055)

**This** film is really **bad**, so **bad** that even Christopher Lee **cannot save it**. A **poor** story an even **poorer script** and just **plain bad** direction makes **this** a truly outstanding horror film, the outstanding **part being** that it is the **only** horror film that i **can** honestly **say** i **would never ever** watch again. **This** garbage make Plan nine **from outerspace** look like **oscar material**.

### Text Analytics (Microsoft)

**Original Result** : Negative (1.0)     **Adversarial Example Result** : Positive (0.58)

This film is really **bad**, so **bad** that even Christopher Lee cannot save it. A **poor** story an even **poorer** script and just **plain bad** direction makes this a truly outstanding horror film, the outstanding part being that it is the only horror film that i can honestly say i would **never** ever watch again. This **garbage** make Plan nine from outerspace look like oscar material.

Figure 1: The homoglyph adversarial examples for sentiment analysis web services (i.e., Amazon comprehend, Google cloud natural language AI, IBM Watson natural language understanding, and Microsoft text analytics) generated by HYPOCRITE

## 3.1 HYPOCRITE OVERVIEW

In this subsection, the proposed HYPOCRITE for generating homoglyph adversarial examples is described. The key idea of the HYPOCRITE is to replace English characters with other similar international characters (i.e., homoglyph) in order to give dataset noise. Such dataset noise can cause misclassification different from the original results. This is because the original meaning disappears due to the noise. With this key idea, HYPOCRITE can generate homoglyph adversarial examples by appropriately changing text from a word unit to a paragraph through a black-box attacks for natural language web services. The homoglyph adversarial examples mean text that looks the same to the human, but causes different results through AI services. Figure 1 shows the four homoglyph adversarial examples for sentiment analysis web services (i.e., Amazon comprehend, Google cloud natural language AI, IBM Watson natural language understanding, and Microsoft text analytics) generated by HYPOCRITE. As homoglyph adversarial examples, characters with red color and bold font mean homoglyph that looks the same to our eyes but has different Unicode values. As shown in Figure 1, although they look like the same text to perception, the result of the sentiment analysis web service is different from positive to negative, respectively. The user study on the perception of human for adversarial examples is explained in detail in Section 4.2.2, and the score for each MLaaS company's sentiment analysis result is explained in detail in Section 4.1.

## 3.2 ADVERSARIAL EXAMPLE GENERATION

**Algorithm 1:** Non-Targeted Adversarial Example Generation

---

$result_{text} \leftarrow$ get_sentiment($target\_model$, $text$);
$units \leftarrow [word, sentence, paragraph]$;
**for** $unit \in units$ **do**
   $tokens \leftarrow$ tokenize($result_{text}, unit$);
   **for** $token \in tokens$ **do**
      $result_{token} \leftarrow$ get_sentiment($target\_model$, $token$);
      **if** *min_sentiment($result_{token}$) is not original sentiment* **then**
         Append $result_{token}$ to $attackers$;
      **end**
   **end**
   $attackers \leftarrow$ Sort($attackers$) according to descending of original sentiment;
   **for** $result_{token} \in attackers$ **do**
      $AE \leftarrow$ make_adversarial_example($result_{text}$, $result_{token}$);
      $result_{AE} \leftarrow$ get_sentiment($target\_model, AE$);
      **if** *sentiment of $result_{AE}$ is changed* **then**
         return $AE$;
      **else**
         **if** *original sentiment score of $result_{AE}$ is less than original sentiment score of $result_{text}$* **then**
            $result_{text} \leftarrow result_{AE}$;
         **end**
      **end**
   **end**
**end**

**Algorithm 2:** Targeted Adversarial Example Generation

---

$result_{text} \leftarrow$ get_sentiment($target\_model$, $text$);
$unitss \leftarrow [word, sentence, paragraph]$;
**for** $unit \in units$ **do**
   $tokens \leftarrow$ tokenize($result_{text}, unit$);
   **for** $token \in tokens$ **do**
      $result_{token} \leftarrow$ get_sentiment($target\_model$, $token$);
      **if** *max_sentiment($result_{token}$) is not target sentiment* **then**
         Append $result_{token}$ to $attackers$;
      **end**
   **end**
   $attackers \leftarrow$ Sort($attackers$) according to ascending of target sentiment;
   **for** $result_{token} \in attackers$ **do**
      $AE \leftarrow$ make_adversarial_example($result_{text}$, $result_{token}$);
      $result_{AE} \leftarrow$ get_sentiment($target\_model, AE$);
      **if** *sentiment of $result_{AE}$ is target sentiment* **then**
         return $AE$;
      **else**
         $diff_{target} \leftarrow$ target sentiment score of $result_{AE}$ − target sentiment score of $result_{text}$;
         $diff_{max} \leftarrow$ max sentiment score of $result_{AE}$ − max sentiment score of $result_{text}$;
         **if** $diff_{target}$ *is more than* 0 *and* $diff_{target}$ *is more than* $diff_{max}$ **then**
            $result_{text} \leftarrow result_{AE}$;
         **end**
      **end**
   **end**
**end**

This subsection describes the process of our HYPOCRITE that generates homoglyph adversarial examples for sentiment analysis web services in the physical world. The adversarial example generation is classified into a non-targeted adversarial example generation and a targeted adversarial example generation. The non-targeted adversarial example generation means to generate homoglyph adversarial examples that misclassify a positive sentiment into sentiments other than the positive sentiment, or a negative sentiment into sentiments other than the negative sentiment. The targeted adversarial example generation means to generate homoglyph adversarial examples that misclassify a positive sentiment into target sentiments (e.g., negative, neutral, and mixed), or a negative sentiment into target sentiments (e.g., positive, neutral, and mixed). Algorithms 1 and 2 show an adversarial example generation algorithm of the HYPOCRITE. As shown in Algorithms 1 and 2, for the adversarial example generation, HYPOCRITE consists of five steps: (i) Get score of original text (i) Tokenize the text, (iii) Get score of the tokens and sorting, (iv) Make adversarial examples,

and (v) Verify the effectiveness of the adversarial examples. Through this process, the non-targeted and targeted adversarial examples are generated, respectively.

# 4 EXPERIMENTS

## 4.1 EXPERIMENTAL-SETUP

**Dataset** We used IMDB Dataset to evaluate our attacks on sentimental analysis (Maas et al. (2011)). IMDB contains 25,000 reviews for each positive and negative label, respectively. In our experiments, to determine the effect of text length on the success of the adversarial example, the dataset used in the experiment was divided into seven sections by length. The first section is 500 characters or less, and the second section is 500 characters or more and 800 characters or less. The third section is 800 characters or more and 1,100 characters or less, and the fourth section is 1,100 characters or more and 1,400 characters or less. The fifth section is 1,400 characters or more and 1,700 characters or less, and the sixth section is 1,700 characters or more and 2,100 characters or less. The last seventh sections are over 2,100 characters. Then, we randomly sampled 96 review data per divided section. The number of sample reviews (96 reviews) in our evaluation was determined by a statistically recommended sample size when the confidential level is 95%, the population size is 50,000, and the margin of error is 10%.

**Targeted Models**

To evaluate our attacks, we performed the attacks on four sentimental analysis services (Amazon, Google, IBM, and Microsoft) in real world. Since every service had different sentimental labels and scoring systems, we briefly described labeling and scoring metric for each system in Table 1.

Amazon and Microsoft use four sentimental labels (Positive, Negative, Neutral, and Mixed), and the systems provide each score for all the labels. In addition, the final output as the result is the label which is the biggest score among the sentimental labels.

|  | Label | Score metric | Decision |
|---|---|---|---|
| **Amazon** | Positive, Negative, Neutral, Mixed | 0 <Score <1 (For each label) | Label (The biggest score) |
| **Google** | - | -1 <Score <1 (Total score) | Score |
| **IBM** | Positive, Negative, Neutral | -1 <Score <1 (Total score) | Score |
| **Microsoft** | Positive, Negative, Neutral, Mixed | 0 <Score <1 (For each label) | Label (The biggest score) |

Table 1: Four targeted models. Label stands for labels provided by each system, Score metric for sentimental score used by the system, and Decision for final output provided by the system.

On the other hand, IBM uses three sentimental labels (i.e., Positive, Negative, and Neutral) and Google dose not provide any exact label. Both systems provide overall score of the input texts as final output ranging from -1 to 1 (The score closer to -1 means negative, and to 1 means positive). As there is no sentimental decision provided from the systems, we considered the score less than 0 as negative, and more than 0 as positive.

**Baseline** To compare the performance of our attack, we evaluate our attack with an attack which is the most similar method to ours (Wolff & Wolff (2020)). In (Wolff & Wolff (2020)), the attack is targeted to the neural text detectors. Therefore, we performed the baseline attack to our targeted systems and measure the performance of the attack.

**Evaluation Metrics** In order to evaluate the algorithmic performance, we adopt two metrics such as attack success rate and perturbed rate. The average attack success rate is defined as the ratio of generated adversarial examples that cause misclassification to randomly selected samples. In order to show diversity, we evaluated the performance under various misclassification conditions such as non-targeted misclassification and targeted misclassification. The perturbed rate is defined as the ratio of replaced characters to the total characters throughout the sentence.

## 4.2 RESULTS

This section contains a summary of the performance of a targeted attack and non-targeted attack through the black-box attack against four different platforms such as Amazon, Google, IBM, and

| Web Service Platform | Source | Destination | Attack Model | | | |
|---|---|---|---|---|---|---|
| | | | HYPOCRITE | | HOMOGLYPH | |
| | | | Success Rate (%) | Perturbed Rate (%) | Success Rate (%) | Perturbed Rate(%) |
| Amazon | Positive | Non-targeted | 100 | 5.64 | 68.75 | 15.52 |
| | | Negative | 86.46 | 6.61 | 27.23 | 15.37 |
| | | Neutral | 92.26 | 17.62 | 37.05 | 15.51 |
| | | Mixed | 59.67 | 13.41 | 7.29 | 15.09 |
| | Negative | Non-targeted | 94.2 | 13.05 | 27.08 | 15.22 |
| | | Positive | 47.47 | 11.41 | 2.98 | 12.6 |
| | | Neutral | 81.75 | 17.98 | 18.15 | 15.07 |
| | | Mixed | 60.71 | 10.65 | 2.83 | 13.14 |
| Google | Positive | Non-targeted | 99.40 | 7.24 | 78.27 | 6.62 |
| | | Negative | 81.85 | 14.52 | 24.11 | 8.43 |
| | | Neutral | 99.40 | 5.92 | 77.98 | 6.63 |
| | Negative | Non-targeted | 100.00 | 5.26 | 55.65 | 6.75 |
| | | Positive | 94.49 | 15.64 | 6.70 | 8.26 |
| | | Neutral | 99.85 | 5.92 | 55.80 | 6.68 |
| IBM | Positive | Non-targeted | 99.55 | 8.81 | 61.90 | 15.33 |
| | | Negative | 98.81 | 8.21 | 58.04 | 15.38 |
| | | Neutral | 22.77 | 47.62 | 4.32 | 13.82 |
| | Negative | Non-targeted | 98.22 | 11.78 | 7.44 | 13.23 |
| | | Positive | 98.21 | 11.40 | 5.51 | 12.98 |
| | | Neutral | 22.02 | 46.24 | 1.34 | 10.68 |
| Microsoft | Positive | Non-targeted | 97.77 | 12.84 | 64.22 | 15.34 |
| | | Negative | 64.22 | 15.34 | 18.89 | 15.27 |
| | | Neutral | 92.90 | 36.77 | 21.49 | 15.68 |
| | | Mixed | 80.79 | 2.59 | 21.75 | 14.91 |
| | Negative | Non-targeted | 99.85 | 5.36 | 33.78 | 15.28 |
| | | Positive | 96.43 | 21.73 | 3.57 | 10.82 |
| | | Neutral | 92.28 | 36.96 | 4.91 | 12.89 |
| | | Mixed | 88.69 | 3.87 | 25.89 | 14.92 |

Table 2: Overall performance evaluation of our attack model and baseline attack model

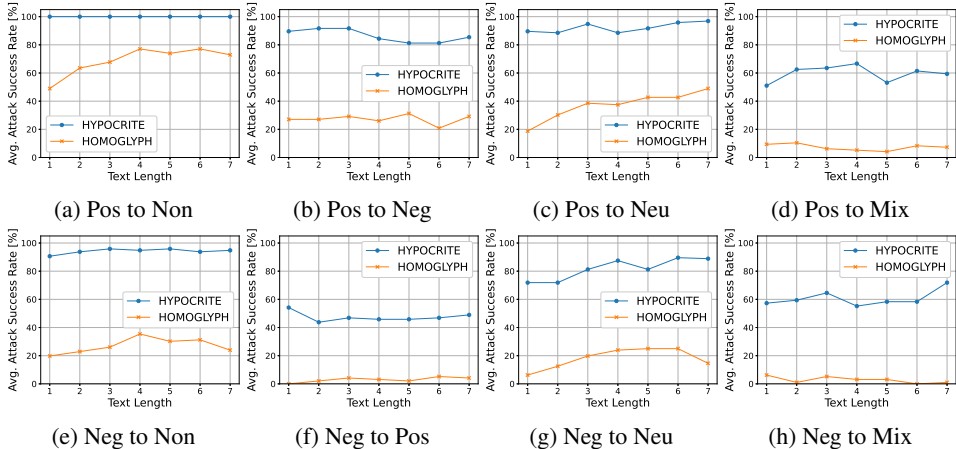

(a) Pos to Non    (b) Pos to Neg    (c) Pos to Neu    (d) Pos to Mix

(e) Neg to Non    (f) Neg to Pos    (g) Neg to Neu    (h) Neg to Mix

Figure 2: The impact of the Amazon web service platform according to non-targeted and targeted adversarial attacks

Microsoft. Table 2 shows the successful attack rate and perturbed rate of each attack. Broadly, our attack is divided into two types such as targeted attack and non-targeted attack. Non-targeted attack means misclassifying the original sentiment classification into another classification other than the existing one. On the other hand, the targeted attack is to perturb a positive or negative classification so that it may be perceived as the specific destination. As shown in Table 2, although the perturbation rates are similar, HYPOCRITE generally performs higher than the baseline (i.e., HOMOGLYPH). We will describe the impact of Web service platform in the next subsection in detail.

### 4.2.1 THE IMPACT OF WEB SERVICE PLATFORM

**Amazon** This section demonstrates the performance of adversary success rate for Amazon. As shown in Figure 2 shown, HYPOCRITE generally shows better performance than HOMOGLYPH on target and non-target attacks. To be detailed, on positive to non (i.e., non-targeted attack) attack,

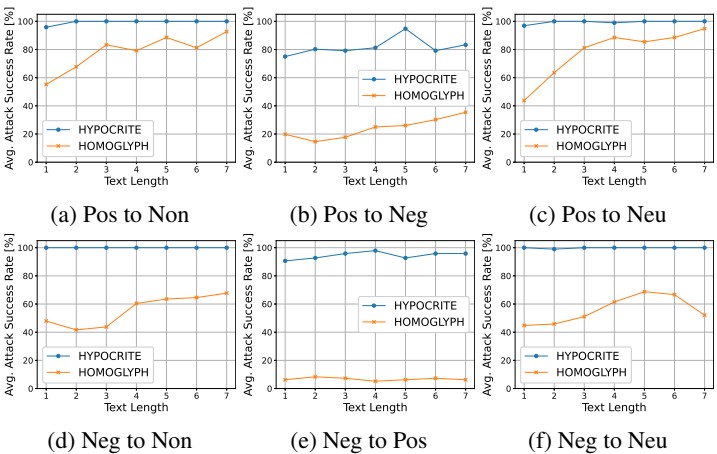

Figure 3: The impact of the Google web service platform according to non-targeted and targeted adversarial attacks

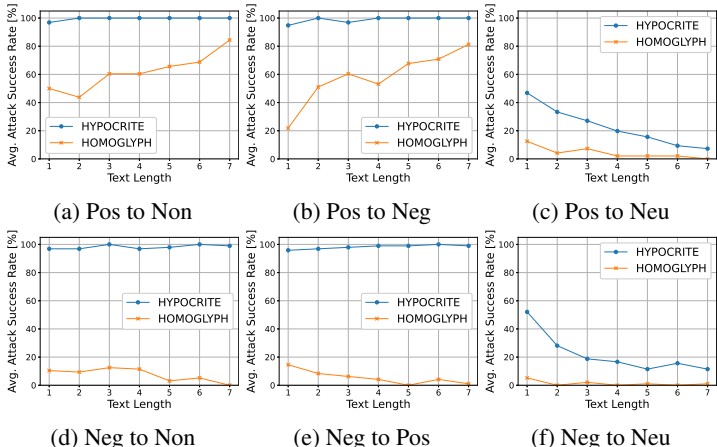

Figure 4: The impact of the IBM web service platform according to non-targeted and targeted adversarial attacks

when HOMOGLYPH performed up to 68.75%, HYPOCRITE performed 100%. Both models' performances of negative to non are lower than positive to non. When HOMOGLYPH performed 27.08 percent, HYPOCRITE still performed a superior success rate of 94.20 percent. In targeted attack cases, except for negative to positive, positive to mix, and negative to mix, comparing with HOMOGLYPH showed less than 40 percent, HYPOCRITE performed significantly high performance. Both models are having low performance on negative to positive, positive to mix, negative to mix but HYPOCRITE is showing significantly higher performance than HOMOGLYPH about 20 times difference. Both models' impact changes belonging to length dependency were hardly be found.

**Google** This section consists of performances of attacks against Google. Figure 3 shows the impact of the text length in Google web service platform. As shown in Fig. 3, since Google API does not provide for targeted attacks with results of mixed, there are only two scenarios for each targeted attack (positive or negative to opposite and neutral). In case of non-targeted attack, HYPOCRITE is showing utmost performance with approximate 100% performance and in contrast, HOMOGLYPH in Google is showing about performance of 66.96%. In targeted attack case, similar to non-targeted attack case, HYPOCRITE also shows good performance in target attack scenarios. Although HYPOCRITE appears relatively low performance in positive to negative, it still performs 2 to 3 times higher than HOMOGLYPH. Lastly, within the text length perspective, HOMOGLYPH performed decreased with length, HYPOCRITE shows consistently high performance regardless of length.

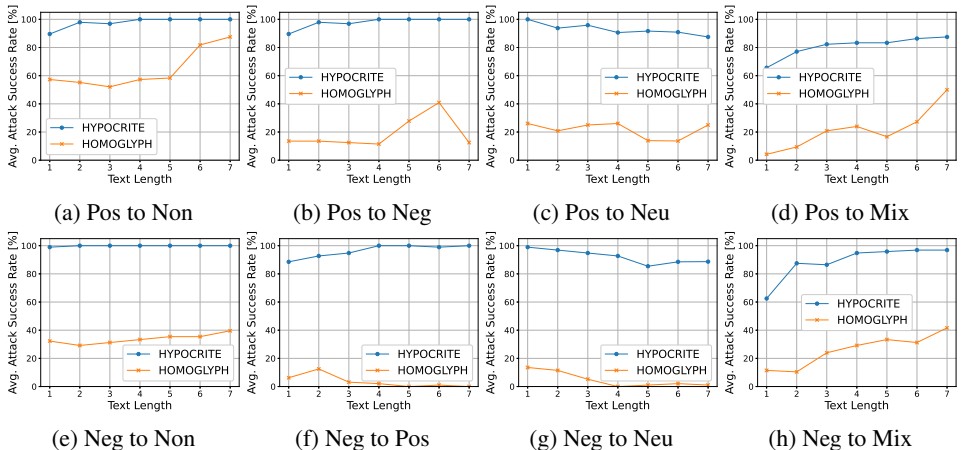

(a) Pos to Non      (b) Pos to Neg      (c) Pos to Neu      (d) Pos to Mix

(e) Neg to Non      (f) Neg to Pos      (g) Neg to Neu      (h) Neg to Mix

Figure 5: The impact of the Microsoft web service platform according to non-targeted and targeted adversarial attacks

**IBM** This section shows another performance against IBM through figure 4. As shown in Figure 4, there is a significant different between two model in non-targeted attack. While HOMOGLYPH is showing intermediate performance in positive to non, it shows fairly low performance. In contrast, HYPOCRITE shows fairly high performance in both scenarios of non-targeted attack. In targeted attack case, unusual result can be found. There were no significant differences compared to other platforms within the result of positive to negative and negative to positive that HYPOCRITE demonstrated better performance than HOMOGLYPH. Yet, unlike other web service platforms, both HYPOCRITE and HOMOGLYPH show poor performance in positive to neutral and negative to neutral. Indeed, during our experiments, the original text was not classified as neutral by IBM. Nevertheless, the baseline showed a result close to 0%, whereas HYPOCRITE showed an average of 22.3%.

**Microsoft** This section demonstrates the performance of adversary attack against Microsoft. As shown in Figure 5, HYPOCRITE shows eminently better performance than HOMOGLYPH on target and non-target attacks. In fact, on positive to non (i.e., non-attack, while HOMOGLYPH performed up to 64.22 percent, HYPOCRITE performed 100 percent. Even though HOMOGLYPH performed 33.78 percent on negative to non, HYPOCRITE performed 99.85 percent which is near to 100 percent. In targeted attack cases, HYPOCRITE showed generally high success rates that are above 90 percent except positive to mix and negative to mix. Compare to that, HOMOGLYPH's highest performance is only 25.89 percent. Even HYPOCRITE showed less performance on negative to mix and positive to mix, it still performed above 80 percent when HOMOGLYPH only performed around 20 percent. The reason why the performance of the attack targeting mixed is lower than other attacks is that the shorter text length provokes the lower performance therefore, the overall success rate of HYPOCRITE decreases.

### 4.2.2 USER STUDY

We conducted a survey of homoglyph adversarial examples using Amazon Mechanical Turk (Mturk) to evaluate the effectiveness of our attack model, and we asked public users to find the words that are changed by homoglyph.

In addition, in order to investigate the effect of text length on the success of the adversarial attack, the IMDB dataset used in the experiment was divided into 7 groups by length. The survey was conducted with random questions selected from each group.

To calculate the success rate of the adversarial example of our model, we defined the attack success rate by quan-

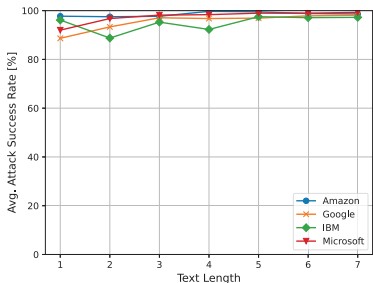

Figure 6: The user study for perception of the homoglyph adversarial examples generated by HYPOCRITE

tifying the number of attack words found by the experimental participants in the given text. Figure 6 shows the specific average attack success rate of expriment participants per platform and the average attack success rate per text length group. Looking at Figure 6, the longer the text is given to the participant, the higher the attack success rate. In particular, the first group had the lowest at 93.63%, and the seventh group with the longest text length was the highest at 98.33%. This shows that the shorter the target text of an adversarial attack, the easier it is for users to recognize the attack.

Through a user study, it was found that the adversarial attack text made by our proposed model was difficult to distinguish with the human eye, so even though it was actually an adversarial attack, it was very difficult for users to recognize.in the user study we experimented with, after informing the user that the word was wrong, we asked the user to look for the wrong or strange word, but in real life (i.e., the situation where the mistake is not told in advance)even if there is a mistake in the word, it is expected that the success rate of the attack targeting the user will be further increased by the word superiority effect (Baron & Thurston (1973)) that unconsciously recognizes the word correctly.

## 5 RESEARCH CHALLENGES

**Effectiveness of perturbed letters in every word** In our work, we perturbed the whole words to generate adversarial examples. However, every single letter in the words could be perturbed with a letter which looks the same but has a different unicode value. From the results in Wolff & Wolff (2020) that replacing all the vowels to generate the adversarial examples was the most effective attack, there might be a significant pattern in letter perturbation of the words to make the adversarial attack more effective. In other words, when the letter which has a significant effect on the success of the attack, the attack will be more effective.

**The financial cost to evaluate the results** To test the performance of our attacks, we used sentiment analysis APIs provided by real world companies. However, every time we checked the results using APIs, the use of APIs should be payed. Therefore, we had to sample a part of the whole dataset, and also could not try the further experiments which deal with perturbing letters to find which letters are important to the score of sentimental analysis.

## 6 CONCLUSION

In this paper, we studied adversarial attacks using homoglyph against natural language web services in the physical world. The experimental results demonstrate that our HYPOCRITE is more effective than other baseline in terms of both attack success rate and perturbed ratio for four popular web service platforms. Also, through the user study, we showed our homoglyph adversarial examples was difficult for users to recognize as a perturbed text. As future work, we will extend the field of our HYPOCRITE to attack to various web services with natural language in addition to sentiment analysis.

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
