# OpenReview forum: "HYPOCRITE: Homoglyph Adversarial Examples for Natural Language Web Services in the Physical World"
_ICLR.cc/2022/Conference — ICLR 2022 Submitted_

### Official Review · Reviewer_JKRL · 2021-11-02

**Correctness:** 4
**Technical Novelty And Significance:** 1
**Empirical Novelty And Significance:** 1
**Recommendation:** 3
**Confidence:** 5

**Main Review:**

Pros:

1. Interesting problem and impressive results.

Cons:

1. Limited evaluation.
2. Incomplete discussion/comparison with prior art, including https://arxiv.org/pdf/2104.05996.pdf




**Summary Of The Paper:**

The authors propose the injection of human imperceptible unicode characters that induce evasion effects in neural based text recognition platforms.

**Summary Of The Review:**

Detailed Comments:

1. The comparison with Boucher et al. is unfair and mostly incomplete. The authors need to better position their work w.r.t prior art to highlight their contributions.
2. Work by Boucher et al. suggests that the proposed attack can be used for a variety of different tasks apart from sentiment detection. Can the authors try the same?
3. The authors do not evaluate any defense methodologies against these attacks. Can the authors comment on the impossibility of defenses? For example, what if one were to convert the string to remove all unicode characters (as a sanitization step)?
4. There’s no evaluation to suggest that the modified strings would fool a human. Was there any MTurk study performed to support these claims?
5. What are some other use cases of the purported attack? Can the authors comment on this?

---

### Official Review · Reviewer_QX9M · 2021-11-02

**Correctness:** 3
**Technical Novelty And Significance:** 1
**Empirical Novelty And Significance:** 2
**Recommendation:** 3
**Confidence:** 3

**Main Review:**

Strengths:
The paper conducts experiments on real-world industry platforms to showcase the performance. The paper shows good empirical results on attack success rate on the APIs.

Weaknesses:
The novelty of the proposed method is not demonstrated very well in the paper. The algorithms described in section 3.2 are basically doing brute force searching with the sorting on the sentiment scores queried from the APIs. It is unclear how much technical contribution this method provides.
In the experimental comparison section, the paper only shows the results on the attack success rate and perturbed rate but does not mention the number of queries used by each method. As the authors said in the paper, the queries to the APIs need to be paid and are expensive, the query number is actually a very important metric to show. My intuition is that since the proposed method is a pretty straightforward search, the query number should be high.

Minor:
“based on the sentiment analysis ?sentiment-analysis) based on the user’s review” on page 1.


**Summary Of The Paper:**

The paper proposes an attack method that generates homoglyph adversarial examples for NLP APIs. The method replaces English characters with other international characters that look similar. The paper shows empirically that the method achieves good performance for several real-world APIs.


**Summary Of The Review:**

The technical novelty of the proposed method is limited. The empirical evaluations are not solid enough to demonstrate significant contributions.

---

### Official Review · Reviewer_y6Cv · 2021-11-02

**Correctness:** 2
**Technical Novelty And Significance:** 2
**Empirical Novelty And Significance:** 2
**Recommendation:** 3
**Confidence:** 4

**Main Review:**

The paper is very poorly written and hard-to-follow, full of grammar and language errors. Asides from this, I'm having hard time understanding the contribution of this work over Boucher et al.'s "Bad‌‌‌‌‌‌‌‌‌‌‌‌‌‌‌‌‌‌‌‌‌‌‌‌‌‌‌‌‌‌‌‌‌‌‌‌‌‌‌‌‌‌‌‌‌‌‌‌‌‌‌‌‌‌‌‌‌‌‌‌‌‌‌‌‌‌‌‌‌‌‌‌‌‌‌‌‌‌‌‌‌‌‌‌‌‌‌‌‌‌‌‌‌‌‌‌‌‌‌‌‌‌‌‌‌‌‌‌‌‌‌‌‌‌‌‌‌‌‌‌‌‌‌‌‌‌‌‌‌‌‌‌‌‌‌‌‌‌‌‌‌‌‌‌‌‌‌‌‌‌‌‌‌‌‌‌‌‌‌‌‌‌‌‌‌‌‌‌‌‌‌‌‌‌‌‌‌‌‌‌‌‌‌‌‌‌‌‌‌‌‌‌‌‌‌‌‌‌‌‌‌‌‌‌‌‌‌‌‌‌‌‌‌‌‌‌‌‌‌‌‌‌‌‌‌‌‌‌‌‌‌‌‌‌‌‌‌‌‌‌‌‌‌‌‌‌‌‌‌‌‌‌‌‌‌‌‌‌‌‌‌‌‌‌‌‌‌‌‌‌‌‌‌‌‌‌‌‌‌‌‌‌‌‌‌‌‌‌‌‌‌‌‌‌‌‌‌‌‌‌‌‌‌‌‌‌‌‌‌‌‌‌‌‌‌‌‌‌‌‌‌‌‌‌‌‌‌‌‌‌‌‌‌‌‌‌‌‌‌‌‌‌‌‌‌‌‌‌‌‌‌‌‌‌‌‌‌‌‌‌‌‌‌‌‌‌‌‌‌‌‌‌‌‌‌‌‌‌‌‌‌‌‌‌‌‌‌‌‌‌‌‌‌‌‌‌‌‌‌‌‌‌‌‌‌‌‌‌‌‌‌‌‌‌‌‌‌‌‌‌‌‌‌‌‌‌‌‌‌‌‌‌‌‌‌‌‌‌‌‌‌‌‌‌‌‌‌‌‌‌‌‌‌‌‌‌‌‌‌‌‌‌‌‌‌‌‌‌‌‌‌‌‌‌‌‌‌‌‌‌‌‌‌‌‌‌‌‌‌‌‌‌‌‌‌‌‌‌‌‌‌‌‌‌‌‌‌‌‌‌‌‌‌‌‌‌‌‌‌‌‌‌‌‌‌‌‌‌‌‌‌‌‌‌‌‌‌‌‌‌‌‌‌‌‌‌‌‌‌‌‌‌‌‌‌‌‌‌‌‌‌‌‌‌‌‌‌‌‌‌‌‌‌‌‌‌‌‌‌‌‌‌‌‌‌‌‌‌‌‌‌‌‌‌‌‌‌‌‌‌‌‌‌‌‌‌‌‌‌‌‌‌‌‌‌‌‌‌‌‌‌‌‌‌‌‌‌‌‌‌‌‌‌‌‌‌‌‌‌‌‌‌‌‌‌‌‌‌‌‌‌‌‌‌‌‌‌‌‌‌‌‌‌‌‌‌‌‌‌‌‌‌‌‌‌‌‌‌‌‌‌‌‌‌‌‌‌‌‌‌‌‌‌‌‌‌‌‌‌‌‌‌‌‌‌‌‌‌‌‌‌‌‌‌‌‌‌‌‌‌‌‌‌‌‌‌‌‌‌‌‌‌‌‌‌‌‌‌‌‌‌‌‌‌‌‌‌‌‌‌‌‌‌‌‌‌‌‌‌‌‌‌‌‌‌‌‌‌‌‌‌‌‌‌‌‌‌‌‌‌‌‌‌‌‌‌‌‌‌‌‌‌‌‌‌‌‌‌‌‌‌‌‌‌‌‌‌‌‌‌‌‌‌‌‌‌‌‌‌‌‌‌‌‌‌‌‌‌‌‌‌‌‌‌‌‌‌‌‌‌‌‌‌‌‌‌‌‌‌‌‌‌‌‌‌‌‌‌‌‌‌‌‌‌‌‌‌‌‌‌‌‌‌‌‌‌‌‌‌‌‌‌‌‌‌‌‌‌‌‌‌‌‌‌‌‌‌‌‌‌‌‌‌‌‌‌‌‌‌‌‌‌‌‌‌‌‌‌‌‌‌‌‌‌‌‌‌‌‌‌‌‌‌‌‌‌‌‌‌‌‌‌‌‌‌‌‌‌‌‌‌‌‌‌‌‌‌‌‌‌‌‌‌‌‌‌‌‌‌‌‌‌‌‌‌‌‌‌‌‌‌‌‌‌‌‌‌‌‌‌‌‌‌‌‌‌‌‌‌‌ Characters: Imperceptible NLP Attacks" paper, which has been around for more than 4 months now. Boucher et al.'s paper both investigate homoglyph attacks and more (like the invisible characters) with better depth and they also propose some easy countermeasures against these attacks (which would prevent the proposed attacks).

**Summary Of The Paper:**

This paper proposes using homoglyphs to attack commercial NLP models for sentiment classification. Homoglyphs look like the characters in English language but they are encoded differently and, therefore, treated differently by the model. They experiment on various MLaaS models and show that their attack can effectively cause both targeted and untargeted misclassifications.

**Summary Of The Review:**

- The contributions over "Bad‌‌‌‌‌‌‌‌‌‌‌‌‌‌‌‌‌‌‌‌‌‌‌‌‌‌‌‌‌‌‌‌‌‌‌‌‌‌‌‌‌‌‌‌‌‌‌‌‌‌‌‌‌‌‌‌‌‌‌‌‌‌‌‌‌‌‌‌‌‌‌‌‌‌‌‌‌‌‌‌‌‌‌‌‌‌‌‌‌‌‌‌‌‌‌‌‌‌‌‌‌‌‌‌‌‌‌‌‌‌‌‌‌‌‌‌‌‌‌‌‌‌‌‌‌‌‌‌‌‌‌‌‌‌‌‌‌‌‌‌‌‌‌‌‌‌‌‌‌‌‌‌‌‌‌‌‌‌‌‌‌‌‌‌‌‌‌‌‌‌‌‌‌‌‌‌‌‌‌‌‌‌‌‌‌‌‌‌‌‌‌‌‌‌‌‌‌‌‌‌‌‌‌‌‌‌‌‌‌‌‌‌‌‌‌‌‌‌‌‌‌‌‌‌‌‌‌‌‌‌‌‌‌‌‌‌‌‌‌‌‌‌‌‌‌‌‌‌‌‌‌‌‌‌‌‌‌‌‌‌‌‌‌‌‌‌‌‌‌‌‌‌‌‌‌‌‌‌‌‌‌‌‌‌‌‌‌‌‌‌‌‌‌‌‌‌‌‌‌‌‌‌‌‌‌‌‌‌‌‌‌‌‌‌‌‌‌‌‌‌‌‌‌‌‌‌‌‌‌‌‌‌‌‌‌‌‌‌‌‌‌‌‌‌‌‌‌‌‌‌‌‌‌‌‌‌‌‌‌‌‌‌‌‌‌‌‌‌‌‌‌‌‌‌‌‌‌‌‌‌‌‌‌‌‌‌‌‌‌‌‌‌‌‌‌‌‌‌‌‌‌‌‌‌‌‌‌‌‌‌‌‌‌‌‌‌‌‌‌‌‌‌‌‌‌‌‌‌‌‌‌‌‌‌‌‌‌‌‌‌‌‌‌‌‌‌‌‌‌‌‌‌‌‌‌‌‌‌‌‌‌‌‌‌‌‌‌‌‌‌‌‌‌‌‌‌‌‌‌‌‌‌‌‌‌‌‌‌‌‌‌‌‌‌‌‌‌‌‌‌‌‌‌‌‌‌‌‌‌‌‌‌‌‌‌‌‌‌‌‌‌‌‌‌‌‌‌‌‌‌‌‌‌‌‌‌‌‌‌‌‌‌‌‌‌‌‌‌‌‌‌‌‌‌‌‌‌‌‌‌‌‌‌‌‌‌‌‌‌‌‌‌‌‌‌‌‌‌‌‌‌‌‌‌‌‌‌‌‌‌‌‌‌‌‌‌‌‌‌‌‌‌‌‌‌‌‌‌‌‌‌‌‌‌‌‌‌‌‌‌‌‌‌‌‌‌‌‌‌‌‌‌‌‌‌‌‌‌‌‌‌‌‌‌‌‌‌‌‌‌‌‌‌‌‌‌‌‌‌‌‌‌‌‌‌‌‌‌‌‌‌‌‌‌‌‌‌‌‌‌‌‌‌‌‌‌‌‌‌‌‌‌‌‌‌‌‌‌‌‌‌‌‌‌‌‌‌‌‌‌‌‌‌‌‌‌‌‌‌‌‌‌‌‌‌‌‌‌‌‌‌‌‌‌‌‌‌‌‌‌‌‌‌‌‌‌‌‌‌‌‌‌‌‌‌‌‌‌‌‌‌‌‌‌‌‌‌‌‌‌‌‌‌‌‌‌‌‌‌‌‌‌‌‌‌‌‌‌‌‌‌‌‌‌‌‌‌‌‌‌‌‌‌‌‌‌‌‌‌‌‌‌‌‌‌‌‌‌‌‌‌‌‌‌‌‌‌‌‌‌‌‌‌‌‌‌‌‌‌‌‌‌‌‌‌‌‌‌‌‌‌‌‌‌‌‌‌‌‌‌‌‌‌‌‌‌‌‌‌‌‌‌‌‌‌‌‌‌‌‌‌‌‌‌‌‌‌‌‌‌‌‌‌‌‌‌‌‌‌‌‌‌‌‌‌‌‌‌‌‌‌‌‌‌‌‌‌‌‌‌‌‌‌‌‌‌‌‌‌‌‌‌‌‌‌‌‌‌‌‌‌‌‌‌‌‌‌‌‌‌‌‌‌‌‌‌‌‌‌‌‌‌‌‌‌‌‌‌‌‌‌‌‌‌‌‌‌‌‌‌‌‌‌‌‌‌‌‌‌‌‌‌‌‌‌‌‌‌‌‌ Characters: Imperceptible NLP Attacks" paper is not clear at all, the authors cite this work but have not evaluated as a baseline or discussed the differences of their work.

- Poorly written, makes reviewing painful, I would recommend revising, reorganizing and proofreading. For example, the sentence in the abstract, "This  hypocrisy can disrupt normal AI services provided by the cloud companies.", here, I have never seen the word "hypocrisy" instead of the word "attack", it seems like an effort to paraphrase or use translation software from another language.

- No mention of easy and trivial countermeasures against this attack.

- Evaluating transferability of the attacks between different services would be useful. The authors complained about the API costs, what if the attacker crafts the adversarial example against a local sentiment classification model and submits it to one of the APIs? Currently, it is not clear how much each adversarial example costs and judging by the algorithm, it might be restrictive.

---

### Decision · Program_Chairs · 2022-01-20

**Decision:**

Reject

**Comment:**

The paper presents attacks against sentiment recognition NLP systems using specially crafted homoglyphs. The novelty of the proposed method is, however, marginal; contributions over some of the related work are unclear. Furthermore, the quality of writing is insufficient. The authors provided no response to reviewers' comments.